# Withdrawal Interval Estimation of Doxycycline in Yellow Catfish (*Pelteobagrus fulvidraco*) Using an LC-MS/MS Method Based upon QuEChERS Sampling Preparation

**DOI:** 10.3390/foods10112554

**Published:** 2021-10-23

**Authors:** Ning Xu, Bo Cheng, Miao Li, Zhoumeng Lin, Xiaohui Ai

**Affiliations:** 1Yangtze River Fisheries Research Institute, Chinese Academy of Fishery Sciences, Wuhan 430223, China; xuning@yfi.ac.cn; 2Hu Bei Province Engineering and Technology Research Center of Aquatic Product Quality and Safety, Wuhan 430223, China; 3Aquatic Products Quality and Standard Research Center, Chinese Academy of Fishery Sciences, Beijing 100141, China; 4Institute of Computational Comparative Medicine (ICCM), Department of Anatomy and Physiology, College of Veterinary Medicine, Kansas State University, Manhattan, KS 66506, USA; miaoli@ksu.edu (M.L.); zhoumeng@ksu.edu (Z.L.)

**Keywords:** doxycycline, LC-MS/MS, QuEChERs, withdrawal time, yellow catfish

## Abstract

Limited information on residue depletion of doxycycline (DC) is available in yellow catfish (*Pelteobagrus fulvidraco*) using the determination of LC-MS/MS based upon a rapid and simple method of sample preparation. This study collected plasma and tissue samples of yellow catfish at pre-determined time points following 3-day consecutive oral administration of DC at 20 mg/kg. The samples were prepared using a QuEChERS (Quick, Easy, Cheap, Effective, Rugged, and Safe) approach. The limit of detection and the limit of quantitation were 2.5 and 5 µg/kg or µg/L, respectively, for DC in plasma and tissues (e.g., muscle + skin, liver, kidney, and gill). The recoveries of DC ranged from 67.2% to 86.2%. The decision limit (CCα) and the detection capability (CCβ) were from 106.2 to 127.8 μg/kg or μg/L. The withdrawal times of DC in muscle + skin were estimated to be 22 d based on the guidelines in China and Europe and 27 d based on Japan’s standard. Overall, this study not only provides an efficient method to rapidly determine the DC concentrations in fish-derived tissues but also provides important information on the safety assessment of DC in aquatic animal-derived food products.

## 1. Introduction

To date, it is well accepted that the use of antibiotics is still among the most efficacious approaches in global aquaculture for prophylactic and therapeutic diseases of fish caused by various pathogens, such as *Edwardsiella ictalurid*, *Aeromonas hydrophila*, *Streptococcus*, *Vibrio*, etc. [1]. Although antibiotics reduce the fish mortality rate to elevate fish production and economic profit, they also bring a serious threat to food safety for humans due to the accumulation of drug residues in the fish bodies that can be delivered into the human body via the food chain. Excessive drug residues may produce various toxicological effects in humans, including vomiting, diarrhea, allergies, and hormone-like reactions [2]. It can even potentially cause an increase of drug resistance in pathogens [3]. To guarantee food safety, most countries stipulate the maximum residue limits (MRL) for approved drugs in edible tissues of food animals, including pigs, chickens, cattle, fish, etc. When meat products are available in markets, drug residue levels must be below MRLs to ensure food safety and minimize the negative impacts for humans.

Doxycycline (DC), a well-known drug of tetracyclines in human, veterinary, and aquaculture medicine, can broadly oppose various bacteria and other pathogens, comprising chlamydiae, mycoplasmas, protozoan parasites, and rickettsiae [4]. It has been approved in Japan, India, China, and the Philippines to use for curing various fish diseases [5]. In China, the label dose is 20 mg/kg/day for 3–5 days in fish [6]. Based on the information citing no observed adverse effect level (NOAEL) and drug resistance of DC, China and the EU have set an MRL of DC at 100 µg/kg in fish edible tissue of muscle + skin [7]. A lower MRL was stipulated in Japan as 50 µg/kg in muscle + skin of Perciformes, such as horse mackerel, bonito, sea bass, mackerel, sea bream, and tuna [8]. In the USA, DC is not approved to use in aquatic animals, and the government authorities require that DC must be less than the limit of detection of analytical equipment in imported aquatic products from Vietnam, Malaysia, Indonesia, and South American countries [9]. However, since MRL has been established and applied in practical aquaculture production, available residue data in fish products provided by the official routine monitoring programs have shown that DC is sometimes over the MRL to threaten human health, which is mainly caused by lack of withdrawal time (WT) information in off-label species or illegal use of DC that requires a prolonged WT in edible tissues. Accordingly, from the perspective of food safety, it is necessary to regulate the specific withdrawal interval in target animals.

Yellow catfish (*Pelteobagrus fulvidraco*) is one fish species of *Siluriformes*, which is an omnivorous fish extensively distributed in Asia. Their foraging activities are generally carried out at night, and their food includes small fish, shrimp, various terrestrial and insects, small mollusks, and other aquatic invertebrates. It has strong adaptability to the environment and can live in adverse environmental conditions [10]. Since yellow catfish possess high edible and nutritional values, they are popular aquatic food products, especially in China, Japan, South Korea, and East and South Asia [11]. The increased demands require enough yellow catfish to be supplied in the market for the consumption of humans. For instance, in China, the total cultured production in 2017 reached 480,032 tons and was increased by 10.5% compared to that in 2016 [12]. To further boost the production of yellow catfish, the intensive culture method has been extensively used in the culture of yellow catfish. These systems not only bring a high production but also result in many issues, such as causing physiological stress of fish, the deterioration of water quality, and the rapid spreading of bacteria and viruses. Therefore, fish diseases are liable to outbreak when shifting temperature, poor nutrition, and mismanagement. Among others, a serious disease of *Edwardsiellosis*, caused by *Edwardsiella ictalurid*, often leads to large mortality of yellow catfish due to deep tissue and brain infections [13]. Fortunately, since DC has fine penetrability and permeability in tissues, it can pass through various biological membrane barriers to arrive in deep tissues so that it can kill the pathogens. However, the drug residue is also simultaneously accumulated in fish products. To erase the potential concern of food safety of DC-treated fish, the WT of DC in yellow catfish is urgently needed.

Many pharmacokinetic studies of DC have been reported in most of the veterinary animals, including pigs [14], chickens [15], calves [16], and horses [17], and a few aquatic animals, such as tilapia [18], grass carp [19], and channel catfish [20]. The tissue residue depletion of DC has also been investigated in pigs [21], hens [22], broilers [23], cattle [24], and grass carp [25]. However, limited information on the residue depletion kinetics of DC is available in yellow catfish. Besides, most existing studies used HPLC or LC-MS/MS to determine DC concentrations in animal tissues [26,27,28,29,30]. In general, the limit of detection of LC-MS/MS is far lower than that of HPLC, which can satisfy the ever-increasingly stringent standards. Thus, LC-MS/MS determination for DC is a wise choice despite its high cost. Additionally, the extraction of tetracyclines from complex matrices has always been a challenging and time-consuming work because tetracyclines can readily combine with bivalent metal ions, especially iron and magnesium ions, in animal-derived tissues. To enhance the extraction efficiency, a mild acid buffer of 0.1 M EDTA-MacIIvaine solvent (pH = 4.0) is generally used to bind the bivalent metal ions in target tissues, thereby facilitating the extraction of tetracyclines. However, the traditional extraction method using a buffer is still complicated, time-consuming, and expensive, as the extractant needs to be further cleaned up by solid-phase extraction, such as HLB. In order to further save time and cost, we recently developed a QuEChERS (Quick, Easy, Cheap, Effective, Rugged, and Safe) method to determine DC’s residue concentrations in aquatic animals [31]. Based on this new method and considering the importance of WT of DC in yellow catfish, this study aimed to use an LC-MS/MS method based on a QuEChERS sample preparation to investigate tissue residue kinetics of DC and evaluate its WT in yellow catfish.

## 2. Materials and Methods

### 2.1. Chemicals, Reagents, and Standards

Zhongbo Aquaculture Biotechnology Co. Ltd. (Wuhan, China) provided DC powder (purity grade ≥ 98%) used for oral administration. The purity grade of DC standard was more than 98%, bought from Dr. Ehrenstorfer GmbH (Augsburg, Germany). The purity grade of DC-D_3_ standard was more than 95%, also bought from Dr. Ehrenstorfer GmbH (Augsburg, Germany). The formic acid, methanol, water, and acetonitrile were all HPLC-grade purchased from J-T Baker (Philipsburg, PA, USA) and Thermo Fisher Scientific (Waltham, MA, USA). Inorganic reagents of citric acid monohydrate, ethylenediaminetetraacetic acid disodium (EDTA-Na_2_), and sodium dihydrogen phosphate were obtained from Shanghai Guoyao Company (Shanghai, China). CNW Technologies (Shanghai, China) provided the Cleanert C_18_ and plastic consumables, such as centrifugal tubes, syringes, and filter membranes.

### 2.2. Standard Solutions

An accurately weighed 10 mg of DC was prudentially dissolved in methanol to get an ultimate level of 400 µg/mL. This solution was used as a standard stock solution. We pipetted 1 mL of the standard stock solution, diluted to 10 µg/mL, and used it as a working standard solution. According to the same method, 1 mg of the labeled internal standard (DC-D_3_) was dissolved in methanol to get a concentration of 100 µg/mL, used as a standard stock solution. One mL of this solution was diluted to 1 µg/mL with methanol as a working standard solution. These solutions were all stored at −20 °C and remade per 1 or 3 months.

### 2.3. Experimental Animals and Management

A total of 100 yellow catfish (110.5 ± 8.5 g) were purchased from the culturing base of the Yangtze River Fisheries Research Institute (Wuhan, China). All fish was reared in tanks (100 × 60 × 80 cm, length × width × depth) receiving water (26 L/min). The various parameters of water quality were kept in appropriate ranges by determining total ammonia nitrogen levels, nitrite nitrogen levels, dissolved oxygen levels, and pH values every day. The water temperature was maintained at 24 ± 0.5 °C by an air-conditioner and an aquarium heater. In the initiation of the experiment, the fish were acclimatized for 14 days with feeding drug-free feed. The ingredients of feed included 76.95% moisture, 12.79% crude protein, 3.83% crude lipid, and 3.36% ash provided by the Nutritional Research Group in our institute. The negative control samples of plasma, liver, kidney, muscle + skin, and gill were collected from drug-free treated fish and preserved at −20 °C, which was used to establish an LC-MS/MS method for DC.

### 2.4. Experimental Design

A suitable amount of DC powder was dissolved into pure water to obtain an ultimate level of 20 mg/mL for oral treatment. The fish was randomly allocated into 11 groups, with 6 fish in each group. Afterward, the fish were weighed and immersed in MS222 solution with a concentration of 50 mg/L until the fish belly was facing up. The DC solution was given to fish by oral gavage at 20 mg/kg using a 1-mL microinjector linked to a plastic hose. After oral gavage, fish may regurgitate the drug into the water environment. Therefore, each fish was inspected to see whether it had regurgitated the drug in the individual tank; if this occurred, the fish was replaced by a treatment-free fish. In compliance with this method, the fish was administered treatment orally once a day for 3 days. All experimental procedures were approved by the Fish Ethics Committee of Yangtze River Fisheries Research Institute, Chinese Academy of Fishery Sciences.

Approximately 1.5 mL of blood was drawn from the caudal vein of six fish for each time point at 0.25, 0.5, 1, 3, 5, 7, 14, 21, 28, 35, and 42 days using a heparinized syringe following 3-day consecutive administrations. Subsequently, tissue samples of liver, muscle + skin, kidney, and gill were also gathered from each fish. Then, plasma samples were obtained by centrifuging blood at 1500× *g* for 4 min. All plasma and tissue samples were preserved at −20 °C up to determination.

### 2.5. Sampling Processing

The processing procedures of samples were performed as a QuEChERs method reported by our previous study [31]. In brief, 1 g (muscle + skin, liver, kidney, and gill) or 1 mL (plasma) of the samples was weighed into a 15-mL tube. A 10 µL of internal standard solution (1 µg/mL) was spiked into each sample. Afterward, 5 mL of McIlvaine buffer (pH = 4) was added to each sample with shaken for 30 s. After standing for 10 min, 5 mL of acetonitrile with 3% formic acid was added to samples. Subsequently, 1 g of NaCl was weighed and added to each mixture for liquid-liquid partitioning. Mixtures were shaken for 30 s, then treated with ultrasound for 5 min, and next centrifuged at 3500× *g* for 5 min. The resulting supernatant was decanted into a 10-mL tube. Subsequently, samples were extracted twice according to the same method described above. The combined supernatant was purified by 200 mg of C_18_ and 1 g of MgSO_4_ power. The mixture was shaken for 30 s and centrifugated at 7000× *g* for 5 min. The extract was pipetted to another 10-mL tube and condensed to completely dry by a gentle nitrogen stream in a water bath at 45 °C. The remaining residue was restituted by 1 mL of acetonitrile and 0.01% formic acid water (10:90). The obtained mixture was further removed impurities by centrifugation at 7000× *g* for 5 min and then filtrated using a 0.22-µm nylon filter. Finally, 10 µL of the sample was injected into LC-MS/MS for analysis.

### 2.6. Instrument Settings and Method Validation

An LC-MS/MS system was employed to analyze all prepared samples, including an auto-sampler, an LC binary pump, and a triple quadrupole mass spectrometer (TSQ Quantum Access MAX, Thermo Fisher, Waltham, MA, USA). Data were obtained and processed by the Thermo Xcalibur software (Copyright 2.1.0). The analyte was separated by a Hypersil Golden (150 mm × 2.1 mm, 3 μm), and the column temperature was set as 30 °C. The mobile phase was separated into A and B. The mobile phase A included 75% methanol, 25% acetonitrile, and 0.1% formic acid. The aqueous phase B is pure water with 0.1% formic acid. The gradient elution procedure with a flow rate of 0.2 mL/min was adopted in this study. The initial percentage of A was 10% for 1 min, elevated to 100% in 5 min, keeping this status for 2 min, and declining to primary percentage in 0.1 min. The injection volume of each sample was set as 10 μL.

The heated electrospray ionization source at positive mode was used to determine the target analyte, and the ion source parameters were optimized by monitoring the MS/MS spectra of the analyte. The corresponding mass spectra parameters of spray voltage, vaporizer temperature, and ion transport tube temperature were set as 3500 V, 350 °C, and 350 °C, respectively. The high-purity nitrogen was used as auxiliary gas with a pressure of 50 arb and sheath gas with pressure at 30 psi. The ultra-high purity argon was conducted as collision gas with a pressure of 1.50 mTorr. Additionally, Q1 peak width, Q3 peak width, and scan time were set as 0.70 amu, 0.50 amu, and 0.1 s, respectively. The collision energy was set as 23 eV, which made DC’s parent ion of *m*/*z* 445 crack to the qualitative ion of *m*/*z* 145. The collision energy was set as 18 eV, which made DC’s parent ion of *m*/*z* 445 crack to the quantitative ion of *m*/*z* 428. Regarding the internal standard of DC-D_3_, the collision energy was set as 17 eV, which made the parent ion of *m*/*z* 448 crack to *m*/*z* 431.

The validity of the LC-MS/MS method was assessed by the analytical guideline of the EU Commission Decision 2002/657/EC. The various indices, including specificity, the limit of detection (LOD), the limit of quantification (LOQ), linearity, decision limits (CCα), the detection capability (CCβ), recovery, and precision, were determined for method validation. LOD was delimited when the concentration could produce a 3-time area to noise baseline. LOQ was delimited when the concentration could produce a 10-time area to noise baseline.

To avoid interferences from evaluating possible exogenous or endogenous factors in samples, we analyzed 10 blank samples from disparate sources by LC-MS/MS. If there was an interference presented in the plot, the sample extraction and purification could be improved to reduce the interference. If the interference was not combined with target compounds, an LC separation condition could be re-developed.

Due to the obvious matrix effect of the determination of DC in fish samples, the standard solution of DC was spiked into plasma and tissues at different levels of 5, 10, 20, 50, 100, and 500 μg/L or μg/kg as the fortified samples. These samples were prepared and determined by the above-mentioned methods. Then, matrix-fortified calibration curves were constructed by regressing realistic concentrations of DC and the ratios of peak area for target chemical to the area of internal standard. Meanwhile, the regressed slope, intercept, and correlation coefficients were also estimated.

Through analyzing 20 blank samples spiked with DC at the maximum residue limits (MRL, 100 µg/kg) in the muscle + skin of fish, the decision limits (CCα) were estimated. Since no regulatory MRL was presented in inedible tissues, the spiked concentration of DC in liver, kidney, plasma, and gill was temporarily set according to that in muscle + skin. The detection capability (CCβ) was evaluated as the decision limit plus 1.64 times the corresponding standard deviation being equal to the standard deviation at the MRL level.

The recovery was evaluated by the determination of spiked blank samples (six replicates for each concentration) at the concentrations of 0.5, 1, and 1.5 times of MRL. By comparing the areas of the fortified concentrations to actual corresponding concentrations, the recoveries were calculated.

The precision of this method was estimated by the repeatability and reproducibility. Fortified samples as described above were analyzed by the same operator and instrument on the same day to calculate the repeatability value. Secondly, fortified samples were daily prepared, processed, and determined on three separate days using the same method with the same instrument and operator to calculate the reproducibility values.

### 2.7. Withdrawal Time Analysis

The WT of DC in plasma and tissues were evaluated by WT 1.4 developed by the European Medicines Agency (EMA). Although the WT is only needed to calculate for the edible tissue (i.e., muscle + skin) according to EU and other regulations, the WT calculations for inedible tissues (i.e., liver, kidney, and gill) were also performed in this study. This was done to gain a comprehensive idea on the time when DC is depleted to be below the safe concentrations in various tissues in yellow catfish. Furthermore, it can provide validation information on the estimation of WTs using a physiologically based pharmacokinetic (PBPK) modeling approach [32].

## 3. Results

### 3.1. Method Validation

Through the analysis of spiked samples by LC-MS/MS, the LOD and LOQ were determined as 2.5 µg/kg and 5 µg/kg in various tissues, respectively. This method has a good specificity without interfering with ions appearing at the same retention time of DC. The different matrix-fortified calibration curves of DC were constructed across concentrations from 5 to 500 μg/kg in plasma, liver, kidney, gill, and muscle + skin, respectively. Good linearities were displayed in calibration curves by high correlation coefficients (r = 0.996–0.999). The spiked samples at different concentrations of 0.5, 1, and 1.5 times of MRL were performed to assess the recoveries in plasma, liver, kidney, gill, and muscle + skin, respectively. Table 1 showed the corresponding results of recovery, repeatability, and reproducibility of DC in spiked samples. In brief, recoveries in plasma and various tissues ranged from 67.2% to 86.2%. The decision limit (CCα) and the detection capability (CCβ) were estimated from 106.2 to 127.8 μg/kg or μg/L. The relative standard deviations of within-day ranged from 2.1% to 4.3%, and relative standard deviations of between-day ranged from 3.5% to 7.4%, which satisfied the requirements of determination.

### 3.2. Residue Depletion and WT Assessment of DC

The residue depletion data in plasma and tissues are shown in Table 2 and Figure 1. The results showed that different concentrations of DC were presented in plasma and various tissues. At the first sampling time, the order of concentrations from high to low was arranged as gill, liver, kidney, plasma, and muscle + skin. The highest levels in plasma and tissues occurred at the first sampling time point (0.25 day) after the final oral dosing, and the mean concentrations were 80942.0 ± 9320.9, 82843.4 ± 7791.7, 15979.0 ± 2589.3, 1398.1 ± 41.9, and 1287.5 ± 107.5 µg/kg in gill, liver, kidney, plasma, and muscle + skin, respectively. In liver and gill, DC residue was detectable for a long time, up to 35 days after oral dosing, with residue concentrations of 20.7 ± 4.6 and 16.4 ± 2.7 µg/kg, respectively, and then their concentrations were below LOQ at 42 days. In kidney and plasma, the residue concentrations were 146.7 ± 54.0 and 9.2 ± 2.1 µg/kg, respectively, at 21 days after oral treatment and were below LOQ after 28 days. In muscle + skin, the DC concentration was detectable until 28 days with DC’s concentration of 20.9 ± 4.1 µg/kg, and it was under LOQ after 35 days.

The residue depletion profiles of DC in plasma and tissues are shown in Figure 1. These time-course profiles exhibited that the decreasing velocity of residue concentrations was rapid, from 0.25 to 7 days after oral dosing. The concentrations were decreased by 99.0%, 94.5%, 90.5%, 70.8%, and 60.5% in gill, liver, kidney, plasma, and muscle + skin, respectively from 0.25 to 7 days after oral dosing. After 14 days, the concentrations were decreased progressively to below MRL (100 µg/kg). At 21 days, the concentrations in muscle + skin, gill, and plasma had been declined to be below 100 µg/kg, which were 43.9 ± 3.2, 98.1 ± 16.8, and 9.2 ± 2.1 µg/kg or µg/L, respectively. At 28 days, the concentrations in the liver and kidney were 66.7 ± 10.1 µg/kg and <LOQ, respectively.

Figure 2 displays the plot of the WT estimations for muscle + skin, liver, kidney, gill, and plasma in yellow catfish using WT 1.4 based on the MRL level of 100 µg/kg in China and Europe. Since only seven data groups were permitted to import into the WT 1.4, accordingly, the data sets from 1 to 35 days were used to selected seven time points for estimation. The selected time points were 1, 3, 5, 7, 14, 21, and 28 days for muscle + skin; 1, 3, 5, 7, 14, 21, and 35 days for liver and gill; and 1, 3, 5, 7, 14, and 21 days for kidney and plasma, respectively. These time points were selected as DC was in the terminal depletion phase during the selected time points in respective tissues. The WT was estimated as 31, 27, 22, 28, and 20 days in liver, kidney, muscle + skin, gill, and plasma, respectively, after setting MRL as 100 µg/kg and considering a 95% percentile with a 95% confidence level according to the regulation of China and Europe. To give our study a broader application, we evaluated the WTs of DC in yellow catfish in accordance with the MRL of 50 µg/kg from the Japanese regulatory authority. WTs were estimated to be 34, 34, 27, 32, and 23 days in liver, kidney, muscle + skin, gill, and plasma, respectively (Figure 3).

## 4. Discussion

In the present study, our LC-MS/MS determination method improved the LOD and LOQ of DC in fish-derived matrix compared to a typical HPLC determination method (i.e., LOQ of 5 vs. 50 µg/kg) [31], which can satisfy most of the requirements in residue measurement in aquatic animals. In addition, a simple QuEChERS method was used in this study. It is well-known that the QuEChERS method conforms to the principle of green analytical chemistry due to moving towards simple, quick, high-throughput procedures without reducing sensitivity, selectivity, detectability, accuracy, precision, reproducibility, and representativeness [33]. It can reduce the usage of organic solvents, minimize the size of samples, simplify and streamline the extraction and purification procedures, and save cost. Moreover, QuEChERS is a highly effective strategy for sample processing, as it is only based on two main steps: (a) the extraction step, based on acetonitrile or other modified reagents by partitioning between an aqueous and an organic layer using NaCl, MgSO_4_, and Na_2_SO_4_; and (b) the clean-up step, based on a dispersive solid-phase involving various cleaners, such as C_18_ sorbent, PSA sorbent, NH_2_ sorbent, Alumina-N sorbent, etc. [34,35]. In light of these strengths, this method has been ubiquitously used in the determination of veterinary drugs, pesticides, mycotoxins, hormones, and persistent organic pollutants [33,35]. Compared to the traditional method [19,23,36], the currently developed method reduced the amount of extraction sample and extraction reagent and reduced the time of sample preparation. In the clean-up step, we adopted a dispersive solid phase to replace solid-phase extraction cartridges, such as HLB, for removing impurities, which greatly reduces the analytical cost. Therefore, this method is a simple and economical approach for determining DC concentration in fish-derived matrices. This method also has some shortcomings. One is that the absolute recovery of DC is low in plasma and tissues of yellow catfish. In the future study, we will optimize and improve this issue.

DC is a widely used antibiotic in veterinary and fishery medicine for therapy and prophylaxis of animal diseases. To avoid its toxicology, many countries have regulated the MRL of DC in various food animals. China established the MRL of DC in the muscle (100 µg/kg), fat 300 (µg/kg), liver (300 µg/kg), and kidney (600 µg/kg) for porcine, bovine, and poultry species and in muscle + skin (100 µg/kg) in fish [37]. Japan has also stipulated MRLs of DC in the muscle (50 µg/kg), fat (50 µg/kg), and liver (50 µg/kg) for chickens and pigs and in muscle + skin (50 µg/kg) in Perciformes, such as horse mackerel, bonito, mackerel, sea bass, tuna, and sea bream [8]. The EMA has also formulated the MRLs in bovine, porcine, poultry, and finfish, and their MRL values are the same as in China [38]. In order to ensure the safety of food products derived from DC-treated fish, it is paramount to establish a reliable and accurate WT of DC in fish. Therefore, this study examined the residue elimination regularities of DC in yellow catfish receiving a label use of consecutive 3-day oral treatments at 20 mg/kg.

We found that DC concentrations were the highest in relevant tissues and plasma at the initial time after multiple oral doses; DC concentrations in the liver were the highest compared to the same time stage in other tissues (except the first sampling time) and presented the slowest depletion, up to 35 d following oral administration; the second highest concentrations were displayed in gill and were still detectable until 35 d as well. These findings are similar to the DC depletion profiles in grass carp receiving the same treatment regimen under the same environmental temperature [25]. The concentration of DC in liver and gill all displayed the highest concentration in both species in comparison to plasma and other tissues. High concentration in the liver may be partly due to DCs existing enterohepatic circulation in the absorption system of animals and humans to result in DCs reabsorption. Relative high concentration presenting in the gills is possible to be attributed to DC’s reabsorption from excreta in water by gill. Subsequently, the concentrations in the liver of yellow catfish at the initial two sampling time points and in the gill of yellow catfish at the first time point were higher than that in liver and gill of grass carp at the same time stage [25]. Additionally, the concentration in liver of grass carp was higher than gill, and the same trend was present in yellow catfish except for the first sampling time point. These differences may be caused by disparate dispositions of DC in both species and different individual operations of drug administration. After that, the elimination rate in yellow catfish became faster than grass carp; hence, the residue concentration was less in yellow catfish. The concentration of DC was decreased to about half of the initial value in liver of yellow catfish at 1–3 d but 3–5 d in liver of grass carp after oral doses. The relevant DC concentration in liver of yellow catfish reached half of the initial concentration at 1–3 d but 3–5 d in gill of yellow catfish after oral treatments.

This study also found that the DC concentrations in yellow catfish in muscle + skin, kidney, and plasma decreased faster than in grass carp [25]. The DC residues in muscle + skin of yellow catfish could be detectable only up to 28 d and in kidney and plasma only until 21 d. However, the corresponding data in grass carp of muscle + skin and kidney were quantifiable up to 42 d and in plasma till 35 d, which were far longer than that in yellow catfish. These discrepancies may be due to many reasons related to the disposition of DC in different fish species. Firstly, the physiological structures between grass carp and yellow catfish display apparent differences. Yellow catfish have a stomach, but grass carp do not have a stomach. Takii and co-authors reported that the absorption, digestion, and excretion rates of tiger puffer (*Takifugu rubripes*) without a stomach were conspicuously slower than red sea bream (*Pagrus major*), which have a stomach, through giving feed containing the indicator of Cr_2_O_3_ [39]. Additionally, the intestine length of carnivorous fish was one-fourth or one-third of the body length, but that of phytophagous fish was 2–5 times the body length. The phytophagous fish possess a slower excretion rate than carnivorous fish to digest complicated carbohydrates, especially cellulose. Based on this view, grass carp, belonging to phytophagous fish, have a longer intestine length. Nevertheless, yellow catfish, belonging to omnivorous fish, have a relatively short intestine length, between that of phytophagous and carnivorous fish. Therefore, it can be speculated that the excretion of yellow catfish for DC is faster than that in grass carp. Secondly, mature yellow catfish are attributable to small-sized fish with a body weight of 50–200 g, whereas mature grass carp pertain to medium-sized fish with a body weight of 5–10 kg. The metabolic rate is commonly described to scale as a power function of body size [40,41]. Paschoal et al. [42] proved that the fish size may influence the metabolism and depletion rate of tetracyclines and that the small-sized fish and lighter fish had a higher metabolism rate for the given drug.

Yellow catfish is a highly commercial and nutritive fish species popularized in Asia. Although many antimicrobial chemicals have been heavily used in their intensive cultivation, the information on residue depletion of those chemicals is scarce. So far, only a few studies have been published by Chinese scientists. It has been reported that the WT of oxytetracycline in yellow catfish was below the MRL at 46 days after 5-day oral administration at a dose of 100 mg/kg at 25 °C, which has a long WT [43]. Yang [44] reported that the WT of florfenicol was at least 3 days in yellow catfish following 3-day oral administrations at a dose of 10 mg/kg, which had a short WT. Some environmental pollutants distributed and eliminated in yellow catfish have also been investigated, and the results showed that the elimination half-lives ranged from 17.3–32.5 h [45], which was shorter than that in this study. These discrepancies in WT values may be due to different chemical properties and regulated MRLs of different drugs. From our findings, the WTs of DC in grass carp were considerably longer than those in yellow catfish at the same given dosage regime. The detailed reasons have been described in the above paragraph. Otherwise, the estimated WTs of DC based on the guideline MRL of 100 µg/kg in China and Europe are shorter than those in accordance with the MRL of 50 µg/kg in Japan. This difference is apparently due to different stipulated MRLs in different countries.

## 5. Conclusions

This study reports plasma and tissue residue elimination characteristics of DC in yellow catfish following a label use of consecutive 3-day oral treatments at a dose of 20 mg/kg at 24 °C using the determination of LC-MS/MS based upon a sampling preparation of QuEChERS. The WT values in plasma and tissues were estimated using the WT 1.4 program made by EMA. The estimated WT values were 22, 31, 27, 28, and 20 d in muscle + skin, liver, kidney, gill, and plasma, respectively, in accordance with the MRL of 100 µg/kg in China and Europe, and the WT values became 27, 24, 24, 32, and 23 d, respectively, in accordance with the MRL of 50 µg/kg in Japan. This study provides WT estimates to help ensure the safety of yellow catfish-derived food products and is beneficial for risk monitoring and assessment of DC use in yellow catfish. In addition, the study reports original plasma and tissue residue data that can help establish a PBPK model to predict the DC tissue residues and WT in yellow catfish. A PBPK model for DC in yellow catfish is warranted to be developed, as it can be extrapolated to estimate WTs following other therapeutic regimens and to other fish species.

## Figures and Tables

**Figure 1 foods-10-02554-f001:**
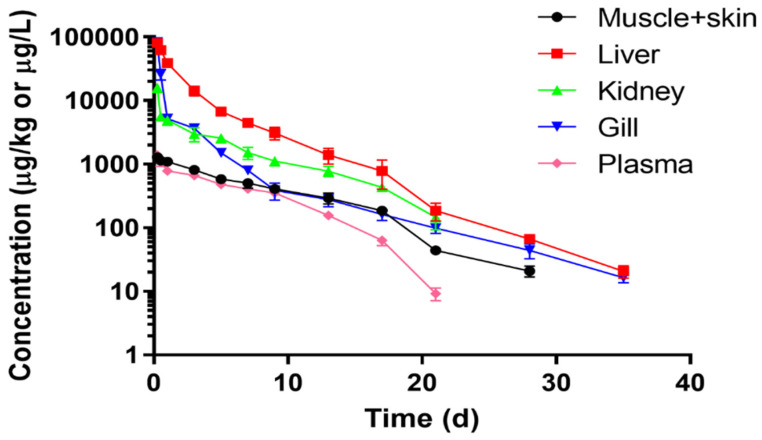
Depletion kinetic curves of doxycycline in plasma and various tissues of yellow catfish (*Pelteobagrus fulvidraco*) after consecutive 3-day oral administrations at 20 mg/kg/day (*n* = 6).

**Figure 2 foods-10-02554-f002:**
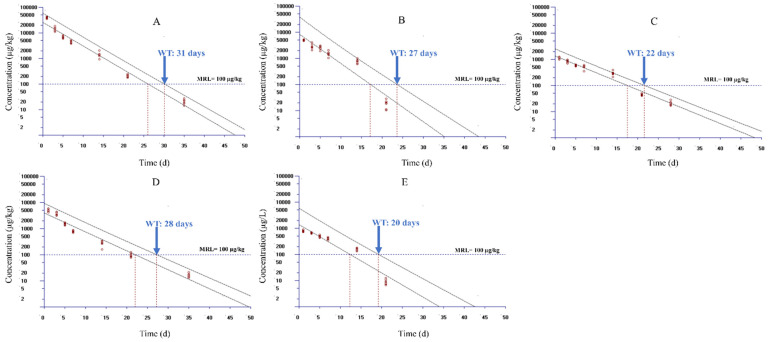
Calculated withdrawal times of doxycycline in yellow catfish (*Pelteobagrus fulvidraco*) following continuous 3-day oral gavage at 20 mg/kg/day based on the MRL standard in Europe and China using the WT 1.4 software. Panels (**A**–**E**) show the results for liver, kidney, muscle + skin, gill, and plasma, respectively. MRL, maximum residue limit for doxycycline from the European Medicines Agency and the Ministry of Agriculture and Rural Affairs of the People’s Republic of China.

**Figure 3 foods-10-02554-f003:**
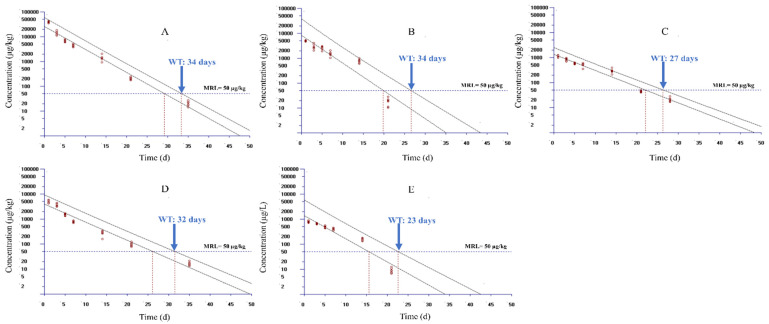
Calculated withdrawal times of doxycycline in yellow catfish (*Pelteobagrus fulvidraco*) following continuous 3-day oral gavage at 20 mg/kg/day based on the MRL standard in Japan using the WT 1.4 software. Panels (**A**–**E**) show the results for liver, kidney, muscle + skin, gill, and plasma, respectively. MRL, maximum residue limit for doxycycline from the Ministry of Agriculture, Forestry and Fisheries of Japan.

**Table 1 foods-10-02554-t001:** The recovery, repeatability, and reproducibility of doxycycline determination in fortified plasma, liver, kidney, gill, and muscle + skin of yellow catfish (*Pelteobagrus fulvidraco*) using LC-MS/MS (*n* = 6).

Tissues	Spiked Concentration (µg/kg or µg/L)	Recovery (%)	Within-Day RSD (%)	Between-Day RSD (%)
Plasma	50	80.2	2.1	3.8
	100	77.3	3.7	5.0
	150	82.1	3.1	4.7
Liver	50	67.2	4.2	6.3
	100	83.7	2.7	5.5
	150	81.9	3.2	4.9
Kidney	50	70.1	4.3	6.0
	100	83.2	2.2	4.5
	150	85.7	4.0	7.4
Gill	50	83.9	3.7	5.8
	100	84.1	4.3	6.7
	150	82.7	3.5	5.2
Muscle + skin	50	80.9	2.9	4.2
	100	86.2	2.6	3.5
	150	84.7	3.0	5.7

RSD, relative standard deviation.

**Table 2 foods-10-02554-t002:** The concentrations of doxycycline in muscle + skin, liver, kidney, gill, and plasma of yellow catfish (*Pelteobagrus fulvidraco*) after consecutive 3-day oral administrations at 20 mg/kg/day (*n* = 6).

Time (days)	Muscle + Skin (µg/kg)	Liver (µg/kg)	Kidney (µg/kg)	Gill (µg/kg)	Plasma (µg/L)
0.25	1287.5 ± 107.5	80,942.0 ± 9320.9	15,979.0 ± 2589.3	82,843.4 ± 7791.7	1398.1 ± 41.9
0.5	1138.2 ± 88.9	62,100.5 ± 8761.3	5622.8 ± 849.7	25,905.2 ± 4864.9	1097.6 ± 171.8
1.0	1090.9 ± 95.3	38,931.5 ± 2580.4	4923.3 ± 849.7	5180.0 ± 646.2	785.6 ± 54.4
3.0	812.6 ± 113.1	14,338.9 ± 2538.2	2984.8 ± 738.5	3655.3 ± 459.8	664.4 ± 30.3
5.0	579.3 ± 33.7	6708.6 ± 767.9	2566.0 ± 403.5	1499.2 ± 123.9	480.0 ± 52.1
7.0	505.1 ± 85.5	4460.5 ± 521.9	1517.8 ± 329.4	794.8 ± 64.3	408.4 ± 34.1
14.0	292.5 ± 56	1389.7 ± 384.6	778.7 ± 140.2	278.7 ± 64.8	156.3 ± 17.7
21.0	43.9 ± 3.2	201.7 ± 26.1	146.7 ± 54.0	98.1 ± 16.8	9.2 ± 2.1
28.0	20.9 ± 4.1	66.7 ± 10.1	<LOQ	43.9 ± 11.7	<LOQ
35.0	<LOQ	20.7 ± 4.6	<LOQ	16.4 ± 2.7	<LOQ
42.0	<LOQ	<LOQ	<LOQ	<LOQ	<LOQ

<LOQ, below limit of quantification.

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
