# Peer review of "Withdrawal Interval Estimation of Doxycycline in Yellow Catfish (Pelteobagrus fulvidraco) Using an LC-MS/MS Method Based upon QuEChERS Sampling Preparation"

_foods, 2021, doi:10.3390/foods10112554_

Round 1

Reviewer 1 Report

Manuscript: Withdrawal Interval Estimation of Doxycycline in Yellow Cat fish

Title: suggestion (Pelteobagrus fulvidraco)

Line 123 – Please rephrase.

The standard stock solution was pipetted 1 mL, and then diluted to 10 μg/mL used as a working standard solution.

Line 172- Subsequently, samples were repeatedly extracted according to the same method described above.

The authors could anticipate the number of times instead of …were repeatedly extracted…

Line 191 - Please revise the sentence: The heated electrospray ionization was used to determine the target analyte using positive mode. and the ion source parameters were optimized by monitoring the MS/MS spectra of the analyte.

Line 238 – The following sentence is too long and difficult to understand, so please rephrase. Although the WT is only needed to calculate for the edible tissue (i.e., muscle + skin) according to EU and other regulations, the WT calculations for inedible tissues (i.e., liver, kidney, and gill) were also performed in this study in order to have a comprehensive idea on the time when DC is depleted to be below the safe concentrations in various tissues in yellow catfish and for future comparisons with estimated WTs using a physiologically based pharmacokinetic (PBPK) modelling approach [32].

Line 249 – Please rephrase this sentence, indicating the meaning of respectively. The matrix-calibration curves were constructed across concentrations of 5 - 500 μg/kg in plasma, liver, kidney, gill, and muscle + skin, respectively.

Line 256 - In brief, recoveries in plasma and various tissues ranged from 70.1% to 86.2%. Please check, it seems that recoveries ranged from 67.2-86.2%.

General comments:

  1. Some aspects such as -Preparation of calibration standards and quality control samples; Sample extraction, Validation, Matrix factor, Calibration curve and linearity, Intra- and inter-day assay accuracy and precision, Recovery, Evaluation of Stability under different matrices and under different conditions and total run time of a sample analysis – could be more detailed and all the results shown in supplementary tables
  2. The advantages and disadvantages could be more precisely indicated.
  3. This manuscript provides a method to determine the DC concentrations in fish-derived tissues, the reviewer considers that would be important a deeper discussion with results published by other authors by using the traditional methodologies and similar methodologies.

Reviewer 2 Report

In this paper, the author developed a rapid and easy-to-use approach to successful quantify doxycycline (DC) in yellow catfish by using QuEChERS and LC-MS/MS analysis. The detection limit and limit of quantitation of DC in fish plasma and tissues were carefully measured. And the withdrawal time of DC were calculated based on the different standards and regulations. In conclusion, this manuscript provided an efficient analytical method for safety assessment of DC in aquatic animal-derived food products.

Questions:

  • In page 6, section 3.3, it would be helpful if the author also provide the concentration value of DC in muscle and skin, just keep the same as what you described for DC in liver, grill, kidney and plasma.
  • For Figure 1, Table 1 and 2, please also mention how many replicates you analyzed for each data point (i.e., n = x) in their titles.
  • In the bottom of page, section 3.1, the recoveries of DC were around 70.1%-86.3%. Based on the data showed in the manuscript, the results were very reproducible and precise. But I'm wondering whether it is possible the accuracy could get any improved. It seemed that a small percentage of DC was lost and the results were a little bit underestimated. It would be much helpful if the author could discuss or comment a bit about this part. For example, discuss the extraction efficiency of QuEChERS and anything could do to improve it, or anything else in the whole process could be optimized to improve the recovery, or any planned experiment or investigation in the future.
